# Study on the Removal of Fluorescent Whitening Agent from Paper-Mill Wastewater Using the Submerged Membrane Bioreactor (SMBR) with Ozone Oxidation Process

**Seunghan Ryu [1], Sanghun Lee [1], Hannah Oh [1], Sanghwa Oh [2], Minsoo Park [3], Jinho Kim [3] and Jaeeun Heo [4,*]**

[1] Korea Dyeing and Finishing Technology Institute (DYETEC), Daegu 41706, Korea;
rsh1007@dyetec.or.kr (S.R.); lsh002@dyetec.or.kr (S.L.); hannah5@dyetec.or.kr (H.O.)

[2] School of Architectural, Civil, Environmental, and Energy Engineering, Kyungpook National University,
Daegu 41706, Korea; shoh@knu.ac.kr

[3] Econity Co. Ltd., Yongin 17162, Korea; minsoo@econity.com (M.P.); stevenk@econity.com (J.K.)

[4] Ecoplus Co. Ltd., Cheonan 31075, Korea

\* Correspondence: ejeheo@nature-plus.kr or kmuweb@gmail.com; Tel.: +82-41-5223903

**Abstract:** In this study, paper-mill wastewater was treated using the Submerged Membrane Bioreactor (SMBR) process. In particular, the ozone oxidation treatment process is applied after SMBR to remove the fluorescent whitening agent, which is a trace pollutant and non-biodegradable. Fluorescent whitening agent concentration was indirectly measured by UV scanning and COD concentration. The concentration of COD before SMBR and ozone oxidation was 449.3 mg/L, and the concentration of treated water was 100.3 mg/$\ell$. The COD removal efficiency of paper-mill wastewater through SMBR and the ozone oxidation process was about 77.68%. The optimized amount of ozone was required for the removal of the fluorescent whitening agent after SMBR was 95 mg·$O_3$/$\ell$ calculated by UV scan results. Additionally, the optimized amount of required ozone to remove COD was calculated to 0.126 mg·COD/mg·$O_3$.

**Keywords:** membrane bioreactor (MBR); ozone oxidation; paper and paper-mill; fluorescent whitening agents; wastewater treatment





## 1. Introduction

In the environmental aspect, when considering the reality that the necessity of acquiring water resource is becoming higher due to the expansion of using water resource in various areas, it is a fact and it is necessary to reuse wastewater that occurs stably in the quantitative aspect. According to the UN, almost half of the population of the whole world will lack be living in areas with a lack of water in 2030 due to global warming. It is also foreseen that the development goals and economic activity will be implemented as risk factors if there is no continuous investment in the water facilities [1].

Moreover, the average precipitation amount during 1 year in South Korea is 1341 mm, which is 880 m more than the global average, but the precipitation amount per 1 person is around 13% of the global average due to the high population [2], and the rain is only focused during June to August where interest has been increasing globally on the technical development of reusing replaceable water resource.

The fiber-dyeing industry and the paper-ill industry are energy and water consumption industries. The fluorescent pigments and fluorescent whitening agent make the product look white and remove the discoloration, playing the role of raising the value of the product. Until now, the pigment and whitening agent have been used widely by being applied to various consumables such as paper, detergent, hygiene products, textile, plastic, and paint [3]. It has been reported for an influence on the fluorescent pigments and fluorescent whitening agent that decomposition is not carried out well and that there is a possibility of residue. However, the concentration from a river or lake is not that high

where it has been evaluated to not have significant influence, but there has been an increase in the focus on the exposure of fluorescent pigments and fluorescent whitening agent [4].

Submerged MBR (Membrane Bioreactor) processing can acquire outstanding water quality where it is a focused processing method in the wastewater treatment among the membrane separation process. Especially, the process applied membrane to the biological wastewater processing of the MBR method with the strengths of high-water quality stability to apply to the present industrial wastewater, small sewer, treated water supply, excrement handling, filtration regeneration, and etc. Membrane bioreactors (MBRs) offer an alternative to treatment by the conventional activated sludge (CAS) process [5]. The domestic water-treatment separation membrane technology is applied to sewer and some industrial facilities of village units when compared to the technology in advanced countries. Therefore, domestic technology can be seen as the initial stage, and it is grasped to have great difference with the advanced countries in the operation and control technology fields of the separation membrane.

In this study, the advanced oxidation process (AOPs) is the water treatment technology to remove the harmful pollutions that cannot be processed with the existing technology due to low biodegradation or high chemical stability [6], and it displays an excellent effect in controlling the slightest number of contaminants. MBR was more effective than conventional activated processes in eliminating common pharmaceuticals and other polar compounds. If compounds cannot be removed through MBR, oxidative processes like ozonation were proposed [7]. Therefore, this study applied the submerged membrane bioreactor system to separate the organic matter included in the wastewater, acquired high-quality effluent water through the SMBR, and evaluated the water quality characteristic on the SMBR treatment high quality. Furthermore, the decomposition characteristics of fluorescent pigments and the water quality characteristics of the paper-mill wastewater were evaluated by introducing the ozonation process on the effluent water, including the fluorescent whitening agent.

## 2. Theoretical Background

### 2.1. Submerged Membrane Bioreactor Process

The Submerged Membrane Bioreactor system is simple, and it has the strength to reduce the power costs. The processing efficiency is very excellent, and the wastewater can be used as treated water [8].

The SMBR process is used in substitution for the final settling tank where complete solid/liquid separation is possible in the final stage, and it has the strength to maintain high concentration for the microorganisms inside the bioreactor. Due to the strengths of reusing treated water, removing organic matters, automation and minimization, and acquisition of treated water quality, the process has been receiving interest. Moreover, many commercialization plants have been composed in the treated water supply, sewage treatment facilities, and advancement of treated water quality. Usually, the normal membrane penetration pressure is operated in below 0.5 bar [9]. Additionally, it is operated in the biological endogenic respiration phase where there is less surplus sludge [10], and the costs consumed for dehydration can be reduced.

### 2.2. Ozone Oxidation

The ozone is a strong oxidizing agent with high oxidation potential (2.08 eVolt) [11], and it creates a quick oxidation reaction with organic and non-organic substances of various forms due to its unique molecular structure. The ozone is very unstable in the water, and it self-decomposes due to cyclic chain reaction to go through middle products such as Hydroperoxide radical, superoxide radical, and ozonide radical to create OH radical with greater reactiveness. The organic matter that exists in the water can be decomposed through the indirect reaction pathway that responds to OH radical and the direct reaction pathway that can directly remove the organic substances. The organic substance forms the ozonide

due to the direct and indirect reaction to be decomposed with the aldehyde and simple organic substances to completely oxidize to water and carbon dioxide [12].

### 2.3. Fluorescent Whitening Agents

Fluorescent pigments and fluorescent whitening agents use oxidation and reducing agents to make the textiles (fiber, paper, pulp, etc.) white. The fluorescent whitening agent treatment is executed because the small portion of yellowish-brown cannot be completely removed in this kind of bleach [13,14]. The cellulose fluorescent whitening agent used in the fiber and paper-mill industry mostly uses diaminostilbene disulfonic acid derivatives. The chemical structure is shown in Figure 1. Stilbene fluorescent whitening agents are used as bistriazinyl derivates of 4,4'-diaminostilbene-2,2'-disulphonic acid, and the soluble fluorescent whitening agent substances are stilbene derivatives [15].

**Figure 1.** Chemical structure of the Stilbene fluorescent whitening agent.

The chemical formula of the fluorescent whitening agent used in this study was $C_{14}H_{14}N_2O_6S_2$. The overall ozonization reaction is described as follows with an example of Stilbene fluorescent whitening agent:

$$C_{14}H_{14}N_2O_6S_2 + 10O_3 \rightarrow 14CO_2 + 4H_2O + 2SO_2 + 2NH_3 \tag{1}$$

## 3. Experiment

### 3.1. Subject Wastewater

The actual wastewater used in this study was the primary chemical treated water from paper-mill Company M in Daegu, Republic of Korea, and the composition is shown in Table 1. The average concentration was calculated from 19 days of measured data (from 16th August to 15th September). Company M's paper-mill waste used the diaminostilbene disulfonic acid derivative used in the paper-mill industry for the composition process. The fluorescent whitening agent of the diaminostilbene disulfonic acid derivative is included in the waste where it was included in the SMBR bioreactor for operation. Based on the leakage number of the SMBR bioreactor, the optimal operation factor of the ozone oxidation was calculated for the high quality of the effluent water through the membrane penetration number through the ozone oxidation, including the small number of diaminostilbene disulfonic acid fluorescent whitening agent.

**Table 1.** Characteristics of paper and paper-mill wastewater.

| Parameter | Concentration Range | Average Concentration |
|:---:|:---:|:---:|
| COD (mg/ℓ) | 314–598 | 449.3 |
| Turbidity (NTU) | 222–567 | 327 |

### 3.2. Experimental Device

The SMBR composition, as seen in Figure 2, is composed of a permeate tank ($0.1 \text{ m}^3$), a submerged membrane aerobic tank ($0.55 \text{ m}^3$), and a back-washing tank ($0.06 \text{ m}^3$). The capacity of the SMBR reactor tank is $5 \text{ m}^3$/day of a pilot plant. To acquire stability of the process operation, 12 min of suction, 3 min of stop, and 15 s of back-washing consecutive

operation methods were adopted. It was installed within the site of Company M's paper-mill waste treatment.

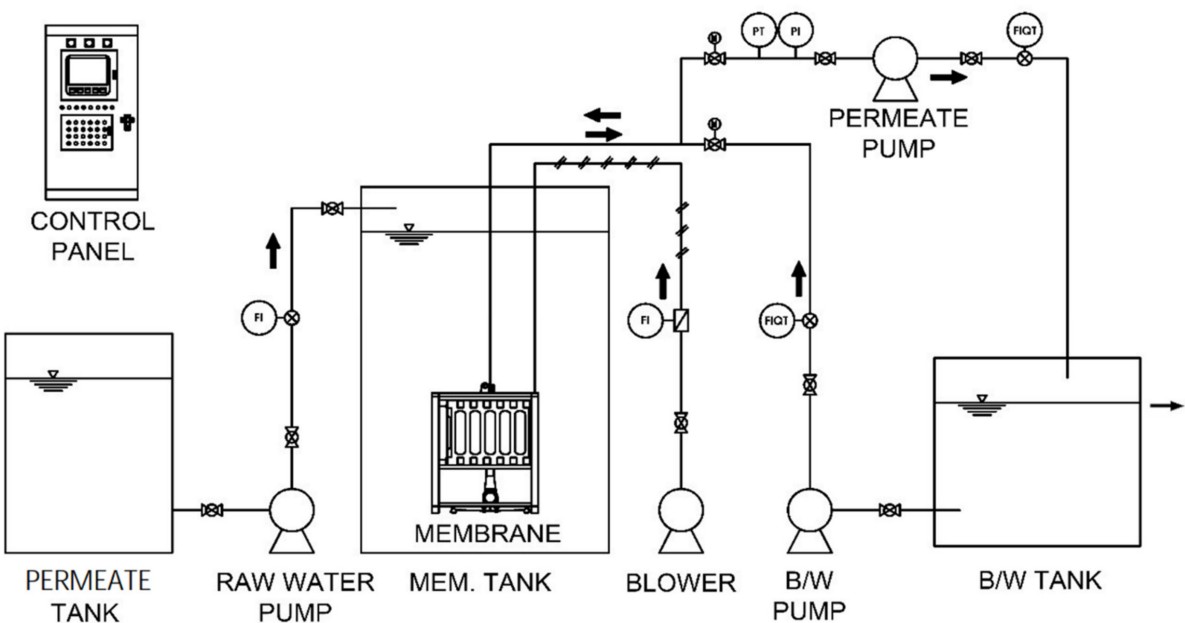

**Figure 2.** A schematic diagram of SMBR system (MEM. TANK: Membrane tank, B/W TANK: Back-washing tank, FI: Flow Indicator, PT: Pressure Transducer, PI: Pressure Indicator, FIQT: Flow Indicator using Quartz).

Generally, the SMBR process operation can produce reuse of high quality even in less than 4 days of SRT (Sludge Retention Time) and 2 h of HRT (Hydraulic Retention Time) [16]. The hydraulic retention time was 4.4 h, and the SRT was 6.6 days for the submerged membrane aerobic used in this experiment. SS meter and drain valve are included in PLC system (Control panel), the emission is conducted if MLSS value was over 3 g/L automatically. The separation membrane used in this research was Company E's submerged fiber membrane (CF-C Type, Yongin, Korea), and the membrane module specification is shown in Table 2. The ozone oxidation device (HIO-600, Yongin, Korea) was used to evaluate the decomposition process on the fluorescent whitening agent. The operation condition is as shown in Table 3. Additionally, for the high quality of effluent water, the ozone oxidation experiment was processed on the SMBR treated water. The ozone oxidation reactor tank is in the structure to maximize contact efficiency of the ozone and the wastewater. The consecutive reactor tank of the ozone contact tank, ozone oxidation reactor tank for stabilization of the residual ozone after contact, and the treatment tank was used, and the structure of the reactor tank is shown in Figure 3.

**Table 2.** Specifications of the membrane module.

| Parameter | Condition |
| --- | --- |
| System Type | Submerged |
| Material | HDPE |
| Membrane Type | Hollow Fiber |
| Pore Size | 0.4 μm |
| Total Membrane Surface Area | 16.8 m$^2$ |

In the case of Advanced Oxidation Process (AOP), 3 mg/L of ozone led to the acquisition of 2 log of the total colon bacillus removal efficiency [17]. By adopting the ozone processing to the high quality of the wastewater reuse, it is judged to help the water quality of reused water and also remove the fluorescent whitening agent.

**Table 3.** Operation conditions of SMBR.

| Parameter | Operation Condition |
|---|---|
| HRT | 4.4 h |
| SRT | 6.6 day |
| Aeration Retention | 25–50 m$^3$/min |
| Dissolved Oxygen | 4.0–5.0 mg/$\ell$ |
| Temperature | $25 \pm 2$ °C |
| pH | 7.0–8.0 |

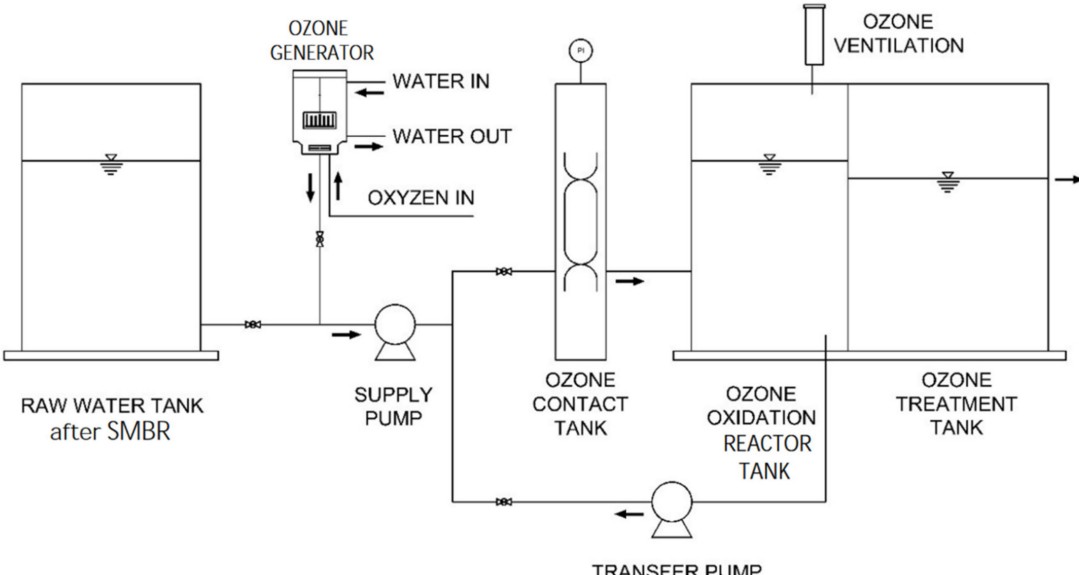

**Figure 3.** Schematic diagram of pilot plant ozone oxidation reactor.

The ozone generator used in the ozone oxidation experiment uses the double derivatives to create high-concentration ozone of high-purity state to apply the ozone generator for 2 $\ell$/min of oxygen flow and ozone concentration of 166 g·$O_3$/m$^3$. The ozone amount used in the experiment was 20.0 g·$O_3$/h. Figure 3 shows the consecutive ozone oxidation reactor used in the experiment. The raw water after SMBR flow went through a strong oxidation process (Totally 130 $\ell$) through treated water from SMBR and ozone contact by being transferred to the ozone contact tank (10 $\ell$). Then, it was moved to the ozone oxidation device (60 $\ell$) to process residual ozone in the wastewater where the non-responsive ozone of the gas is discharged to air after stabilizing from the ozone processor. The treated water after the ozone oxidation reaction was moved to the ozone treatment tank (60 $\ell$), and the circulation process was repeated to control ozone concentration.

*3.3. Measured Items and Analysis*

In this study, COD was analyzed according to the national water standard methods in the Republic of Korea (Method Num.: ES 04315, 1a) [18]. Additionally, TOC (TOC Analyzer, Multi N/C 3100, Analytikjena, Reinach, Switzerland), Turbidity (Turbidimeter, 2100N, Hach, Loveland, CO, USA), MLSS (MLSS Meter, Cosmos-25/B-LineII, Zullig, Rheinech, Switzerland), $UV_{254}$ and $UV_{280}$ scan (UV/Vis spectrophotometer, Cary 8454, Agilent, Santa Clara, CA, USA) were analyzed to evaluate on the SMBR treatment water quality and ozone treatment. The detailed analysis method (TOC: 5310B, Turbidity: 2130B, MLSS: 2540, UV: 5910) of water quality was followed standard methods [19].

## 4. Result and Consideration

The SMBR reactor operation conditions are as below. The aeration strength was 25–50 m$^3$/min, HRT was 4.4 h, and SRT was 6.6 days during the period of the experiment.

It was installed within the waste treatment site of Company M for operation. The primary treated water of Company M was experimented. The HRT was operated as 4.4 h due to factors such as an increase in differential pressure due to concentration of MLSS and inflow of substances within the pump.

The acid radical device to prevent and clean the membrane pollution was in the structure to prevent membrane pollution due to the up-flow air supplied from the acid radical device equipped to the Econity CF-C type cartridge, which was an acid radical device designed to remove the membrane pollution due to the increasing inflow and pressure of the air.

### 4.1. Change of Penetration Flux to the SMBR Bioreactor on the Paper-Mill Wastewater

The change of the penetration flux was reviewed on the membrane module within the SMBR reactor, and it is shown in Figure 4. The cleaning time was set, and the penetration flow was set to 1.5 ℓ/min. Here, the initial differential pressure started at −0.032 bar, and differential pressure increase occurred due to the membrane pollution according to the operation. If the differential pressure was below −0.070 bar, additional aeration and back-washing cycle was increased to maintain −0.032 bar of differential pressure.

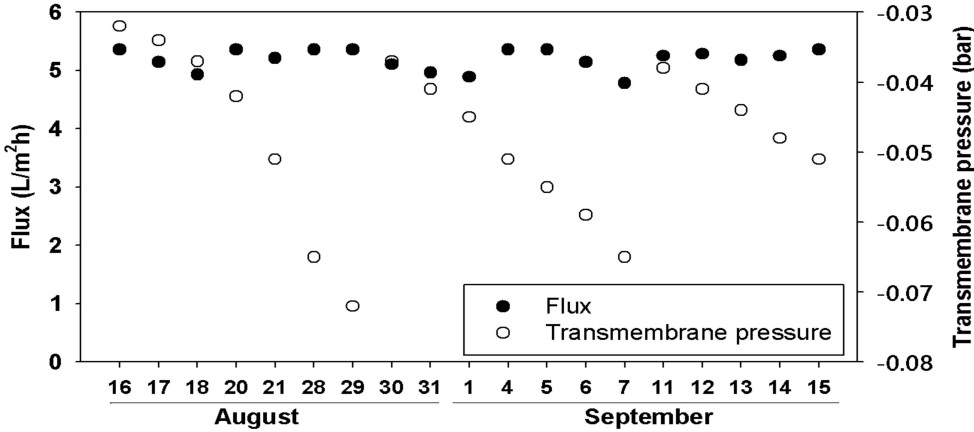

**Figure 4.** The variation of flux during the operating days.

The initial differential pressure was −0.032 bar when setting the flow to 1.5 ℓ/min when starting the research, but there was around 2–8% change in the case of the flow up to 5 days after membrane contamination stated. Here, the change of flux was reduced by 5–8.5%. This means that various subsidiary materials are used in the process of advancing the quality of the product in the paper manufacturing process. Particularly, BOD and COD inducement substances are used with mucoid start or C-stein in the coating process and high-molecular substances such as alum in the line. Due to these substances, they are implemented as factors that influence the membrane flux when operating SMBR. Along with the influence to the membrane penetration performance with a serious change of the lower layer of the membrane surface of the polymer used in the process as the cause of membrane contamination [20], the microorganism proliferation speed increases due to the increase in MLSS within the SMBR reactor tank where it has the same result as the study that reported the increase in microorganisms could become the factor of membrane contamination [21]. Therefore, when applying SMBR in paper-mill wastewater, it is needed to control the back-wash cycle to prevent membrane contamination, not just for satisfying the general wastewater standard.

### 4.2. Change of Organic Contamination on the SMBR Bioreactor on the Paper-Mill Wastewater

4.2.1. Change of MLSS on the SMBR Bioreactor

Figure 5 shows the experiment result of the MLSS concentration change when the HRT was 4.4 h. The MLSS concentration within the SMBR reactor maintained an average of

3026 mg/ℓ. To constantly maintain the MLSS concentration, the PLC program controlled to discharge the concentrated MLSS 2 times a day was used. Due to this, the MLSS within the reactor was maintained constantly to 3026 mg/ℓ, but the SMBR bioreactor's microorganism increase and the control of the pullout amount according to concentration was the core factor to the stable MLSS management to acquire stable treated water quality.

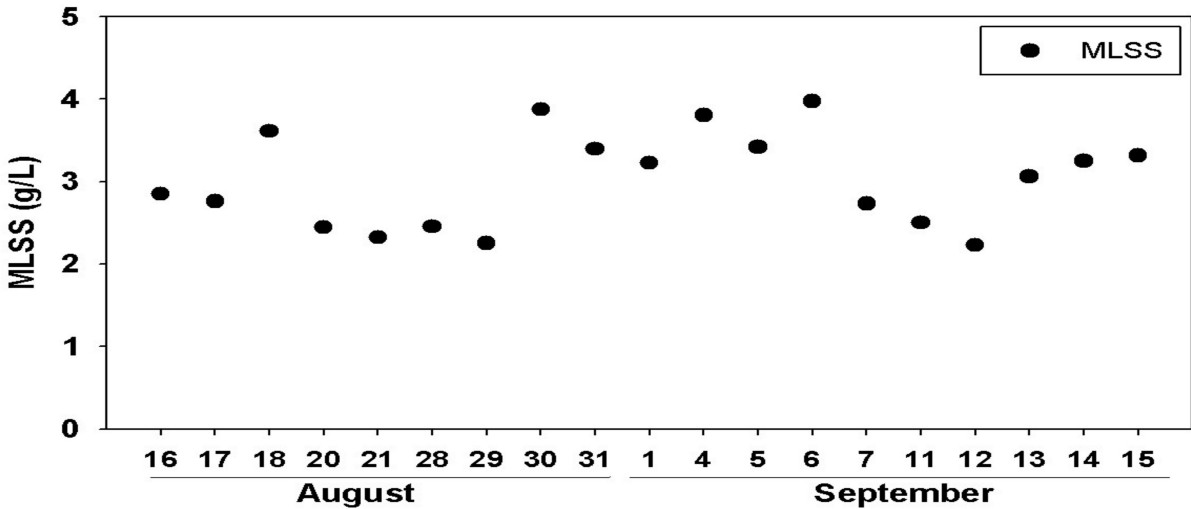

**Figure 5.** Profiles of the MLSS concentration with operating days.

### 4.2.2. Change of Turbidity on the SMBR Bioreactor

The turbidity of the effluent and the influent were analyzed during the period of operation of the SMBR reactor rank. Figure 6 shows the change of turbidity according to the operation of SMBR. The turbidity value range of the influent was 225.0–567.0, where the average turbidity was analyzed to 327 NTU (Nephelometric Turbidity Unit). The turbidity of the SMBR bioreactor penetration water was 0.4–2.1 NTU for the minimum/maximum value, an average of 1.1 NTU. It has appeared that the average was below 3 NTU during the operation of the SMBR reactor tank from the start of the operation. The average turbidity removal rate was above 99%. This means that the large separation was effective due to the submerged fiber membrane reactor.

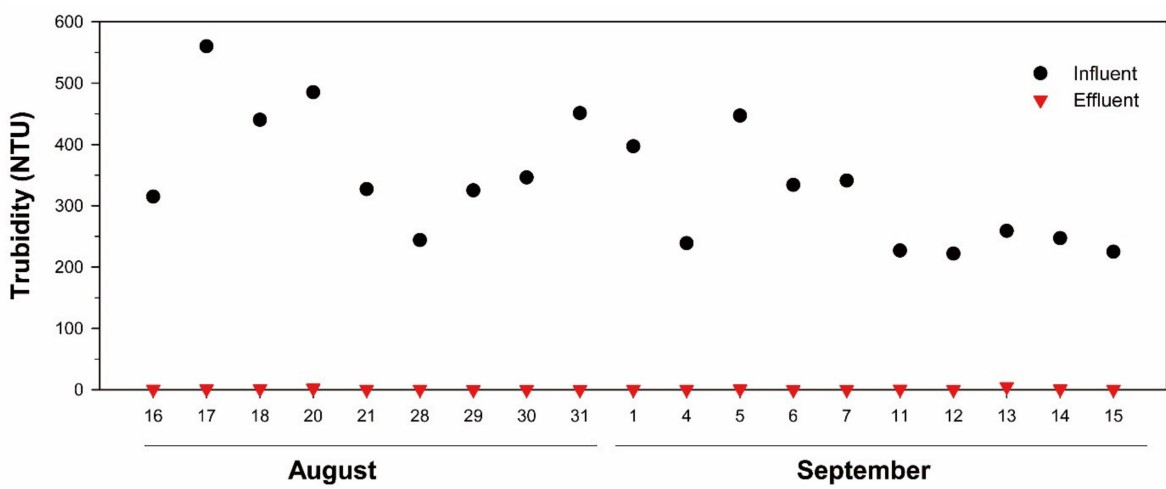

**Figure 6.** Turbidity of paper-mill wastewater and permeate water.

### 4.2.3. Change of COD on the SMBR Bioreactor

Figure 7 is the experiment result that shows the COD concentration change of the SMBR effluent according to the HRT change in the SMBR process. The minimum/maximum value of the SMBR influence was 314–598 mg, and the average COD concentration was analyzed to be 449.3 mg/ℓ. The COD concentration on the water quality that went through biotreatment in the SMBR bioreactor was a minimum of 12–52 mg/ℓ, and the average COD was 100.3 mg/ℓ. Water quality of the influent was too high for 2 days (From 31st August to 1st September). During those days, the company used a lot of starch for the advancement of the paper. When the change of water quality was serious, bubbles would occur where there was difficulty in operating and managing the reactor. Additionally, produced bubbles were influenced the treated water quality, and the COD value of effluent was higher than the effluent without bubbles. However, the microorganisms accommodated from the wastewater treatment in Company M were used for the experiment where the removal efficiency of the COD was maintained at 60.5% (7th September)–89.0% (28th August).

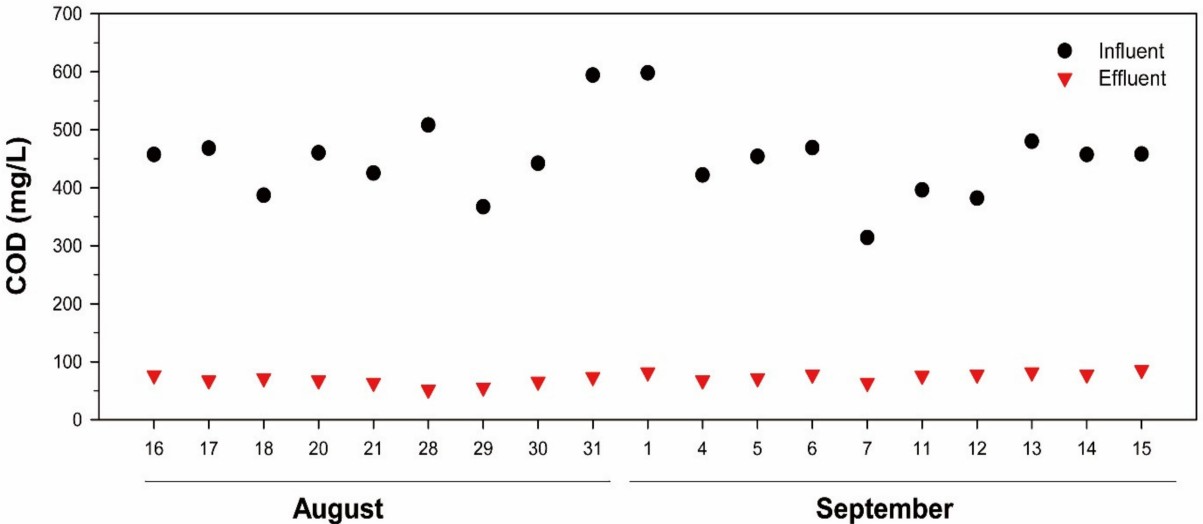

**Figure 7.** Profiles of COD concentration of paper-mill wastewater and permeate water.

### 4.3. Ozone Oxidation Experiment on the Effluent of MBR Bioreactor

#### 4.3.1. Absorption Wavelength Analysis of Ozone Oxidation of MBR Biotreated Water

UV254 Scan analysis method is used as an analysis method of judging the conditions of whether or not a lot of aromatic chemical substance exists on the raw water. In this study, Company M used the diaminostilbene disulfonic acid derivative for manufacturing paper [22], and it has been contained in the water quality of the effluent after biotreatment, where the experiment was conducted based on the effluent from the MBR bioreactor. The diaminostilbene disulfonic acid fluorescent pigment is used often in the paper mill industry and the dye industry, is an aromatic chemical substance oxidized from the p-nitrotoluene, and is a fluorescent whitening agent. The maximum absorption wavelength before oxidization of the diaminostilbene disulfonic acid derivative showed a maximum peak of 280 nm. This reported 280–330 nm or 355 nm for the range of the diaminostilbene disulfonic acid derivative [23]. UV full scanning showed the greatest peak at 280 nm. It reveals that the fluorescent pigment in the paper-mill wastewater contains an aromatic fluorescent dye and various organic chemicals at 280 nm. Reduction of the maximum wavelength was identified after 10 min from the initial starting point of the ozone oxidation. It can be considered that the fluorescent whitening pigment characteristic is lost when 99% of the chromophore gets removed. After 20 min at 280 nm, there was no further change in the UV absorbance. Therefore, the completion for decomposition of the fluorescent whitening agent in the effluent that passed the SMBR bioreactor took 20 min. Ozone dose

is linearly proportional to time for ozone generation. In this study, the velocity of ozone production is 20.0 g·O$_3$/h, as Section 3.2 mentioned. Optimized ozone dose 95 mg·O$_3$/L was calculated through 6.67 g·O$_3$ during 20 min for 70 L volume of raw water.

### 4.3.2. Change of COD due to Ozone Oxidation of SMBR Biotreated Water

The experiment was executed on the ozone oxidation based on the discharged water that passed the SMBR biotreatment. The COD concentration was measured, and the results are shown in Figure 8. The used amount of fluorescent whitening agent in the composition process of the manufacturing process of Company M's paper was calculated to around 7 mg/ℓ when calculating the amount through fluorescent whitening agent use and the total use of COD concentration. The total amount of other pollutants of the effluent showed initial COD of 61.5 mg/ℓ measured every 10 min. The ozone oxidation experiment was executed for a total of 60 min. In the first 10 min of ozone oxidation, the destruction of amino acid of the chromophore occurred as shown in Figure 9. In this serial process, the continuous process of oxidation reduction was judged to be intensively executed. After 20 min of reaction, the oxidation speed relatively slowed down (Figure 9b). After the completion of the reaction, the COD was 14 mg/ℓ. Based on this result and as a result of experimenting with ozone oxidation on the MBR biotreated water, the ozone input needed when connecting to biotreat the paper-mill wastewater was 6.67 g·O$_3$. Here, the change of the intermediate compounds that induce the COD was evaluated by the change of COD and TOC concentrations and resulted in the reaction of removing the sulfone and amino of the fiber or paper and the fluorescent whitening agent within 10 min of the ozone oxidation reaction time in the oxidation experiment of the diaminostilbene disulfonic acid derivative. After rapidly reacting with the ozone, the creation of aldehyde and methyl started [24–26]. From the 20 min point, complete separation of the diaminostilbene sulfonic acid was continued, and from the 30 min point, around 81.5% of the fluorescent whitening agent removal rate was indirectly calculated UV (at absorbance 280 nm) result as shown in Figure 9b. Based on this, 1 mg of ozone is required to remove 0.126 mg of COD of the effluent (from the effluent discharged after SMBR treatment).

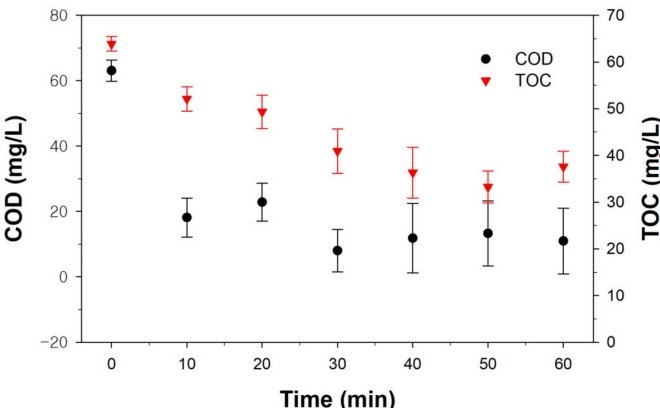

**Figure 8.** The effect of ozone oxidation of the paper and paper-mill wastewater contained fluorescent whitening agents on COD and TOC.

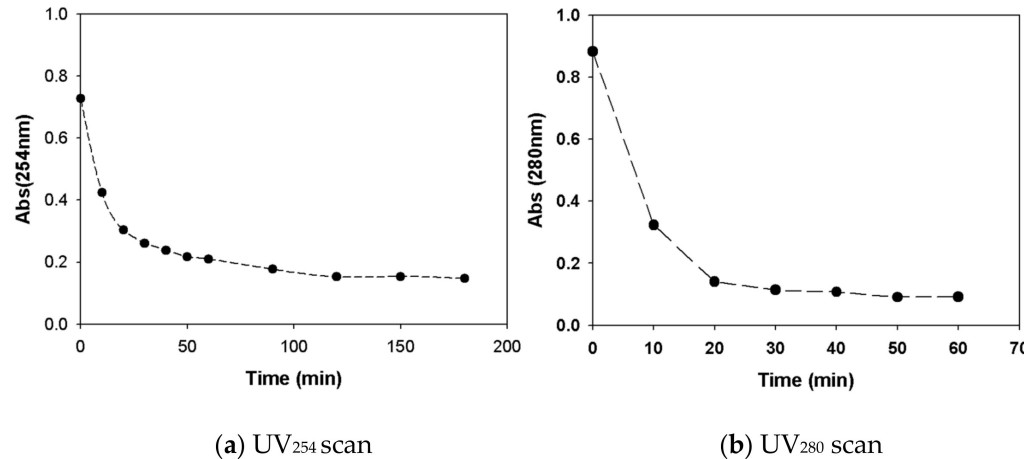

(**a**) UV$_{254}$ scan         (**b**) UV$_{280}$ scan

**Figure 9.** The effect of ozone oxidation of fluorescent whitening agents on UV$_{254}$ and UV$_{280}$ scan.

## 5. Conclusions

As a method to acquire reusable water from the paper-mill wastewater, research by combining submerged membrane bioreactor filtration and ozone oxidation process was executed. The operation condition of the SMBR was MLSS 2.2–4.0 g/L, 25–50 m$^3$/min for aeration strength, 4.4 h for HRT, 6.6 days for SRT, and 1.5 $\ell$/min with differential pressure set to 0.032 bar. The following is the result of executing the ozone oxidation experiment on the filtrated water of the MBR Bioreactor.

The process using the separation membrane is more effective to respond flexibly to the water quality regulation than the conventional type, and it is evaluated to operate the system efficiently. Additionally, to match the water quality regulation that is becoming stricter due to the increase in water consumption due to urbanization and population increase, it is expected that the separation membrane technology will continuously increase.

- The average turbidity of the paper-mill wastewater was 327 NTU, where the turbidity of the filtrated water of the SMBR bioreactor was an average of 1.1 NTU, bringing around 99% of removal efficiency.
- As a result of investigating and analyzing the organic contaminant change of the paper-mill wastewater, the average COD was 449.3 mg/$\ell$ where the COD of the average filtrated water after SMBR biotreatment was 100.3 mg/$\ell$ bringing around 77.68% of removal efficiency.
- The ozone amount needed to remove the fluorescent whitening agent remaining in the filtrated water that passed SMBR of the paper-mill wastewater was 95 mg·O$_3$/L.
- After 20 min, the optimized COD removal was conducted. The optimized ozone amount of SMBR bioreactor was calculated 1 mg of O$_3$ removes 0.126 mg of COD.

**Author Contributions:** Conceptualization, S.R. and J.H.; methodology, S.R.; validation, S.L.; formal analysis, J.H.; investigation, S.O.; data curation, H.O.; writing—original draft preparation, S.R.; writing—review and editing, H.O.; visualization, S.O.; supervision, M.P.; project administration, J.K. All authors have read and agreed to the published version of the manuscript.

**Funding:** This research was funded by Korea Ministry of Environment.

**Institutional Review Board Statement:** Not applicable.

**Informed Consent Statement:** Not applicable.

**Data Availability Statement:** MDPI Research Data Policies.

**Acknowledgments:** This subject is supported by Korea Ministry of Environment as "Global Top Project" (Project No.:2016002210001).

**Conflicts of Interest:** The authors declare no conflict of interest.

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
