# Peer review of "Study on the Removal of Fluorescent Whitening Agent from Paper-Mill Wastewater Using the Submerged Membrane Bioreactor (SMBR) with Ozone Oxidation Process"

_processes, doi:10.3390/pr9061068_

Round 1

Reviewer 1 Report

The paper needs improvement from the very first sentence in the abstract ": In this study, effluent water was produced through Submerged Membrane Bio-Reactor (SMBR) process, which is a simple system and decomposes organic matter contained in wastewater with biological treatment process and performs solid-liquid separation, Especially, ozone oxidation treatment process is applied to effluent water containing fluorescent whitening agent, which is a trace pollutant which is not removed by biological treatment, and influences the quality of reused water." to the very last sentence of the conclusion: "The COD removal amount on the ozone input on the effluent of the SMBR bioreactor was calculated to remove 0.997 mg-COD of 1mg O3. "

Theoretical background is an excess heading where it only contains the added information available in the introduction. "The submerged separation membrane has to withstand the serious shearing force that occurs due to the up-flow of the air and water where very flexible material has to be used. Rather than polysulfone, it is recommended to use flexible materials such as Polyethylene (PE) and Polypropylene (PP)." - This line explains the recommendation (and it is not clear if this fact is based on the present research or the past referenced researches). The recommendations are better mentioned in the conclusion section.

The methodology implemented is hard to grasp. 

The conclusion fails to conclude the main outcome of the research. It just summarizes the findings (the values) that have already been presented in the results and discussion section. The conclusions look stronger if the significance of the findings can be briefed.

Reviewer 2 Report

The publication is interesting and can be published, but in my opinion it should be corrected and supplemented as suggested below.

14

The previous sentence should end before the word "Especially".

20

Instead of 6.67 g-O3/min (per time unit) I propose to enter the amount of ozone per liter of permeate. The amount of ozone fed per unit time only makes sense in comparison with the reactor capacity, which was not given in the summary. Remarks as for line 280

72-73

Is instead of

The processing efficiency is very excellent and the treated water can be used as wastewater[6].

it should not be:

The processing efficiency is very excellent and the wastewater can be used as treated water[6].

77

The membranes used in SMBR are unlikely to remove salt.

80-87

(from the word "However" to 0.5 86 bar [7])

It is not clear from this fragment of the text whether it concerns the research of other authors or the research described in this publication. If these are studies covered in this publication, this section should be moved to Chapter 3, “Experiment”. If, on the other hand, it concerns the research of other authors, I propose to edit this fragment a bit so that there is no doubt.

115

I propose chapter 2.3. supplement with information on how fluorescent whitening agents react with ozone and what products can be formed.

137

Perhaps a period is missing after the word "agent".

140-141

- The description of the devices does not correspond to that in Figure 2.

- Proposes to specify the volume of tanks and the capacity of the installation.

145

- In Figure 2, I suggest changing "B/W Tank" to "permeate tank".

- I propose to supplement Figure 2 with a legend and explain the meanings of the symbols: Fl, PT, PI, FIQT.

152-154

Unclear part of the description. Does the description refer to the operation of SMBR or the ozonation process?

154-159

- I propose to combine this fragment of the text with the text in lines 166-175.

- The descriptions of the devices do not correspond to those in Figure 3.

- What is the capacity of the ozonation plant and the volume of individual tanks?

176

- The device descriptions in Figure 3 do not match the descriptions in the text (lines 152-159)

- Is the "Raw water tank" the same as in figure 2? I suggest adding "Permeate after SMBR"

- Indicate the ozone generator and the place of ozone dosing on the drawing.

- Is the flow direction indicated by the arrow next to "transfer pump" correct?

- Is the description of "Ozone oxidation" correct? Shouldn't there be an "Ozone oxidation reactor tank"?

- Shouldn't “Ozone ventilation” be included in the Ozone treatment tank?

170-175

Unclear description of “pilot plant ozone oxidation reactor” operation.

178-181

How was the concentration of fluorescent whitening agents measured?

184

The MLSS value does not agree with the values given in Table 1. It should probably be g/L.

197

The flow of text and figure 4 are in different units. The text gives the total flow in l / min, while in Figure 4 in m/d. I propose to standardize the method and units of the values given in the text, tables and figures. In my opinion, the flow should be given per m2 of membrane, e.g. L/m2/min. I suggest giving the total capacity in chapter 3, next to the description of the installation, to show the scale of the test bench.

237-238

Fig. 6 does not show that the COD in permeate water was clearly lower than 50 mg / L. The COD values are also consistently lower than 100 mg / L, so why the average value of 100.3 mg / L? If the values given in the diagram are the average values of several measurements, I suggest that this information be included in the description. This also applies to other figures.

247

I propose to move the chapter "Change of MLSS on the SMBR Bioreactor" to the beginning of chapter 4.2, because it contains the operating parameters of the MSBR reactor.

250 and 252

MLSS values do not coincide with Figure 7. Probably the unit is incorrectly given in the text and it should be g/L.

258

The weakest point of the publication is the lack of direct determination of the concentration of fluorescent whitening agents and the fact that only the indirect general parameter of UV absorbance is used.

UV absorbance is a general, non-selective parameter and its size may be determined by many substances contained in wastewater, in particular organic substances. Therefore, on the basis of UV absorbance, both the concentration of fluorescent whitening agents and the effectiveness of their removal cannot be predicted. It allows only an indirect, qualitative evaluation of the process course and this is how the test results should be treated. The work would benefit greatly if the authors supplemented the results with the initial and final concentration of fluorescent whitening agents.

280

The value of 6.67 g-O3/min does not say anything, as the amount of ozone will depend on both the volume of raw water and the concentration of pollutants. To what volume of raw water was ozone administered, 6.67 g-O3/min/L? What was the total ozone dose per L raw water? If the initial concentration of fluorescent whitening agents was known, what was the ozone dose per 1 mg or 1 g of fluorescent whitening agents. The answers to these questions should be included in the publication.

Have studies been performed with other ozone doses? In my opinion, these results should be included in the publication.

299

Remarks similar to the previous ones. To what volume of raw water was ozone administered, 6.67 g-O3/min/L?

What was the total ozone dose per L raw water?

Have studies been performed with other ozone doses?

317

The flow should be calculated per 1 m2 of membranes.

329

Same as for line 280.

Reviewer 3 Report

Manuscript Number: processes-1203746

Study on the Removal of Fluorescent Whitening Agent for Paper-mill Wastewater Reuse using the Submerged Membrane Bioreactor (SMBR) with Ozone Oxidation Process

The present work investigates the performance of combined SMBR-ozone oxidation process for paper-mill wastewater treatment. A pilot plant was used for the treatment of real wastewater, containing a fluorescent whitening agent, for 2 months. Evolution of conventional process parameters (COD, TOC, UV254, MLSS) and membrane fouling are provided. Results revealed a high elimination of organic pollutants.

GENERAL COMMENTS

Investigating new operation strategies for enhancing specific micropollutants removal are on great interesting in MBRs. However, in my opinion, the paper presents important flaws. Experimental procedure (and specifically experimental units) should be explained in detail. Is the ozonation process operation carried out discontinuously? How is the reaction time in the ozonation system regulated? How is the dose of 6.67 g O3/min determined? It is not specified how the physical cleaning procedure of the membrane in the SMBR changes (duration of stop and backwashing cycles, August 29th, and September 7th) when a differential pressure of -0.07 bar is reached. Regarding to the results, in the SMBR there are abrupt increases in MLSS whose explanation has not been detailed (August 18th, August 30th, etc). There is no clear correlation between the evolution of MLSS and fouling (Figs. 4 and 7).There are figures that repeat what is specified in the text, for example, Figures 5 and 6 (there is no appreciable change in COD or in effluent turbidity).

I think the work is interesting and it should be accepted after some major revisions.

SPECIFIC COMMENTS

Sections 1 and 2 can be grouped into a general section called Introduction as specified in the journal's instructions for authors. Section 3 should be called Materials and Methods. Section 4 should be called Results and Discussion.

Abstract

Decimal values for COD should be avoided. From the COD values shown, how is the removal efficiency calculated?

3.Experiment

- (page 4) Table 1: The authors must include mean values ​​and specify values ​​only for the influent of the SMBR. Figures 5 and 6 are redundant with these data. TOC refers to the effluent and therefore must be eliminated from this table. In the same way, the MLSS refer to the suspension of the SMBR and not to the inlet water.

- Fig. 2: The system effluent output stream must be included.

3.2

- (page 4) The text indicates an HRT of 4.4 h and an SRT of 6.6 days, while in Table 3, an SRT of 4.7 d appears.

-Fig 3. (page 5) It is necessary to replace "Raw Water Tank" with "Permeate Tank".

- What is the meaning of "Oxyzen in"? What is the flowrates distribution between "Ozone contact tank" and "Ozone oxidation"?

- What is the pressure value in the "ozone contact tank"?

  1. Results and Consideration

- (page 6) There must be an error in the standard deviation of the MLSS value. There must be an erratum in the SRT value of 6.5 d

4.1.

- Replace "Penetration velocity" with "Permeate flux".

- Replace the value of "penetration flow" of 1.5 l / min by the corresponding value of permeate flow in L / hm2.

- Nothing is specified about TOC removal since influent values are not given.

4.2.3.

- It must be specified how the purge is carried out. The operation of the PLC system should be explained in Materials and Methods.

4.3.1.

- (page 9) The UV254 meter measures at a wavelength of 254 nm, however a spectrum is shown in Figure 8.

4.3.2.

- (page 10) How is the removal efficiency of diaminostilbene sulfonic acid calculated?

- (page 10) How is the dose of ozone calculated?

- (page 10) The standard deviation of the replicates of the tests summarized in Figure 9 should be specified.

  1. Conclusions

The first paragraph includes information on the mode of operation that should not be included in this section. However, in line 317 there must be an erratum in the SRT value of 6.5 d.

Reviewer 4 Report

General comments:

The paper should be re-written with a standard sections of a scientific paper already explained in the guide for author of the MDPI Editorial and Processes journal.

The quality of the paper is low. There is no innovation. The authors evaluate MBR with Ozone but this technology is highly used.

There are mistakes in points and commas. i.e. page 1 line 14 before Especially should be a point instead of a comma.

Specific comments:

Line 19 there is a mistake in re-quired, you mean required?

Line 75. To instead of toe

Section 2.1 can be largely reduced. The authors are explaining the MBR technology which is a consolidated technology largely implemented.

In methodology is expected to see how the pilot plant or facility is equipped. For example, how the authors measured the DO of the mixed liquor? And the temperature. Was there sensors or proves in the reactor? Or the measures were punctual?

Using MBR technology flux refers the ratio between flow and surface area, usually expressed in L/m2·h. In Figure 4 the authors show permeate flux or permeate flow?

Is it relevant to show the days of August or September, why not use the processes day instead dates?

What is the meaning of sentence line 72? The processing efficiency is very excellent and the treated water can be used as wastewater?

Table 1. MLSS (mixed liquor suspended solids) is regarding the biomass not the solid concentration of the wastewater which should be TSS (total suspended solids)

Round 2

Reviewer 1 Report

Comments Response 1 Remarks
General/Abstract: The paper needs improvement from the very first sentence in the abstract ": In this study, effluent water was produced through Submerged Membrane Bio-Reactor (SMBR) process, which is a simple system and decomposes organic matter contained in wastewater with biological treatment process and performs solid-liquid separation, Especially, ozone oxidation treatment process is applied to effluent water containing fluorescent whitening agent, which is a trace pollutant which is not removed by biological treatment, and influences the quality of reused water." to the very last sentence of the conclusion: "The COD removal amount on the ozone input on the effluent of the SMBR bioreactor was calculated to remove 0.997 mg-COD of 1mg O3. "  Response 1: Please check changed abstract. The authors have failed to address the concerns raised in the previous review: It still has flaws from the very beginning). Such as: 1. As per the title, the authors are studying the fluorescent whitening agent for papermill wastewater reuse using a submerged membrane bioreactor (SMBR) with Ozone oxidation process. In other words, this paper studies the mechanism to remove the fluorescent whitening agent for the reuse of the papermill waste water; and this treated wastewater will be reused by the application of the SMBR incorporated with the ozone oxidation process. HOWEVER, the intent is different as per the first line of the abstract, which reads as: the SMBR is a system to decompose organic matter in wastewater through biological treatment process and solid-liquid separation. The SMBR process produces the effluent water. These statements mean two different meanings. 2. Line 15, 16: which is...which is... this sentence needs reforming. If I understood the intent correctly, the authors are trying to inform that the whitening agent cannot be removed by biological treatment, and so it requires ozone oxidation treatment process. 3. Again, the first line in the abstract contradicts with the line 17, 18, which says that the SMBR system in fact reduces the COD in the effluent water (instead of it being an effluent water producer as mentioned in the first line, or a system that makes use of the treated wastewater after removal of the fluorescent whitening agent as claimed in the Title of the paper) . 4. The COD removal efficiency of what??? 6. calculated to be.. 5. The units include "-", for example, mg-O3/L; replace the dash with a dot ".".
Background: Theoretical background is an excess heading where it only contains the added information available in the introduction. "The submerged separation membrane has to withstand the serious shearing force that occurs due to the up-flow of the air and water where very flexible material has to be used. Rather than polysulfone, it is recommended to use flexible materials such as Polyethylene (PE) and Polypropylene (PP)." - This line explains the recommendation (and it is not clear if this fact is based on the present research or the past referenced researches). The recommendations are better mentioned in the conclusion section. Response 2: Thanks. Those are deleted. Still, I find the "2.1 Submerged Membrane Bioreactor process" explains the process extensively (These detailed explanations suit better in the Review papers). The whole content other than the  Line 93-103 can be removed without affecting the usefulness and the intent of the paper. Line 124, 125, 126: the grammar of the sentence.
Methods: The methodology implemented is hard to grasp.  Response 3: Please check revised version of manuscript. Firstly, the authors could have included in the "Response to the Reviewer, what exactly was modified instead of just "Find in the Paper" reply.  1. Again, in line 133, the first line contradicts with the abstract. 2. How was the average concentration in table 1 calculated? 3. How many reps.? 4. What does Mem. Tank, B/W pump in figure 2 mean? 5. the units in table 2 and table 3 (m2, m3/min). 6. Line 183: The treated water completed of...?! GRAMMAR?!  7. Line 191: What are those specific standard methods?! 8. Line 189: How were COD, TOC, Turbidity, MLSS, UV scan done, which instrument, what standards?? 
Results:   1. Line 209: The pressure was set to -0.07 bar and -0.032 bar was maintained, they mean two pressures at the same time?! 2. Line 225: ..rather than the general standard? What does this mean? 3. Line 247: Min/Max value of what?? 249. …above 99%, rather than "more than 99%". 4. Line 259, 260: There were many cases... there were several cases.. These are very general statements. How many specific cases?? 5. Line 263: The water quality was comparatively high... compared with with?! What measured the quality and how was comparison done? 6. Line 263: So.. for the experiment, the authors used the microorganisms collected from the wastewater treatment in Company M?! AND, the removal efficiency throughout the experiment was maintained at a constant level (How?)? (NOT constantly). 7. Line 273: Company M Papermill has been repeated a number of times. Line 282: The greatest peak was seen, OR the greatest peak is shown... GRAMMAR Again.. Line 286: "It can be considered that the fluorescent whitening pigment characteristic is lost when the 99% of the chromophore gets removed"- is what I believed the authors intended to deliver. Line 283: The aromatic chemical substance fluorescent pigment- is this a single phrase that means a pigment OR a chemical?! The phrase is confusing. Line 288: after 20 mins, the oxidation further removed almost all the.....REARRANGEMENT required to make the meaning clear. Line 291: .. is proportional (not proportionate) .. Line 292: There is no chapter here. Line 292: "The complete oxidation time/whitening agent removal time is 20 minutes" is repeated thrice....Line 295: The same 20 mins removal time has been repeated the fourth time. Line 305: What were the organic pollutants detected, identified and how were they measured?? Line 307: Overuse of the company name. It is understood, once the authors mention that the wastewater was collected from the M company, the rest of the paper does contain the samples from the same company, and nowhere else. Line 324: 81.5% of what was removed?? Line 324-326: Rewrite the sentence; the intent is hardly clear. If I understand correctly, the authors are trying to say that 1 mg of ozone is required to remove 0.126 mg COD (of the effluent that is discharged after treatment in the SMBR reactor). Figure 8 (1) and (2), are those Abs and ABS two different things? what is the unit of Abs in the first figure?
FORMATTING   Indentation of "Figure 7:.." and "Figure 6..", 
Point 4: The conclusion fails to conclude the main outcome of the research. It just summarizes the findings (the values) that have already been presented in the results and discussion section. The conclusions look stronger if the significance of the findings can be briefed Response 4: Please check the conclusion. Line 347: Rewrite the sentence. Line 99-103 better suits the conclusion section.
GENERAL COMMENTS:   Please REVISE the manuscript thoroughly. The authors seem to have ignored most of the previous comments. 

Reviewer 3 Report

The authors have improved the previous submission. Main suggested points have been addressed. I suggest this paper should be accepted.

Author Response

Thank you so much for your review.

Reviewer 4 Report

The current version of this paper still presents some weakness. The response to reviews is weak and without scientific evidences. 

There is missing scientific background in the paper and in the sentences concluding some results of this paper. 

I reccommend to reject this paper.

Author Response

Thank you for your review.

Actually, I revised my manuscript again on 7th, Jun.

Could you please check updated version again?

Thank you in advance.

Round 3

Reviewer 1 Report

  • Please address the issues identified in the attached pdf.
  • Furthermore, STILL, the indentations of the lines 228 and 241 do not align.

Author Response

Indentation of Figure 5 is changed.

And these are changed sentences following your review as below.

n this study, COD was analyzed according to the national water standard methods in Republic of Korea(Method Num.: ES 04315, 1a) [25]. 

The detailed analysis method(TOC: 5310B, Turbidity: 2130B, MLSS: 2540, UV: 5910) of water quality was followed Standard Methods[17]. 

UV full scanning is showing the greatest peak at 280 nm. 

After 20 min at 280nm, there was no further change in the UV absorbance.

Ozone dose is linearly proportional to time for ozone generation. 

Reviewer 4 Report

I believe that some of the general references for MBR technology and AOP processes should be updated. The majority of them are past their prime. (<2005).

I propose to include the following references:

MBR technology:

  • MBR technology: Judd, S.J., Judd, C., The MBR Book, (2011).
  • Aeration optimization: 10.1016/j.biortech.2011.12.001

Author Response

Thank you for your review. I added general references as below.

Membrane bioreactors (MBRs) offer an alternative to treatment by the conventional acti-vated sludge (CAS) process [26].

MBR was more effective than conventional activated processes in eliminating common pharmaceuticals and other polar compounds. If compounds cannot be removed through MBR, oxidative processes like ozonation were proposed [27].

  1. S. Judd and C. Judd, "The MBR book: Principles and applications of membrane bioreactors for water and wastewater treatment", 2nd Edition Elsevier, Oxford, UK, 2011.
  2. Bernhard, M., Müller, J. and Knepper, T.P., Biodegradation of persistent polar pollutants in wastewater: Comparison of an optimised lab-scale membrane bioreactor and activated sludge treatment. Wat. Res., 2006, 40, 3419-3428.